# Retweet communities reveal the main sources of hate speech

**Bojan Evkoski**[1,2], **Andraž Pelicon**[1,2], **Igor Mozetič**[1]*, **Nikola Ljubešić**[1,3], **Petra Kralj Novak**[1]

**1** Department of Knowledge Technologies, Jozef Stefan Institute, Ljubljana, Slovenia, **2** Jozef Stefan International Postgraduate School, Ljubljana, Slovenia, **3** Faculty of Information and Communication Sciences, University of Ljubljana, Ljubljana, Slovenia

* igor.mozetic@ijs.si

**Data Availability Statement:** Data availability: The Slovenian Twitter dataset 2018-2020, with retweet links and assigned hate speech class, is available at a public language resource repository CLARIN.SI at https://hdl.handle.net/11356/1423. Model

## Abstract

We address a challenging problem of identifying main sources of hate speech on Twitter. On one hand, we carefully annotate a large set of tweets for hate speech, and deploy advanced deep learning to produce high quality hate speech classification models. On the other hand, we create retweet networks, detect communities and monitor their evolution through time. This combined approach is applied to three years of Slovenian Twitter data. We report a number of interesting results. Hate speech is dominated by offensive tweets, related to political and ideological issues. The share of unacceptable tweets is moderately increasing with time, from the initial 20% to 30% by the end of 2020. Unacceptable tweets are retweeted significantly more often than acceptable tweets. About 60% of unacceptable tweets are produced by a single right-wing community of only moderate size. Institutional Twitter accounts and media accounts post significantly less unacceptable tweets than individual accounts. In fact, the main sources of unacceptable tweets are anonymous accounts, and accounts that were suspended or closed during the years 2018–2020.

## Introduction

Hate speech is threatening individual rights, human dignity and equality, reinforces tensions between social groups, disturbs public peace and public order, and jeopardises peaceful coexistence. Hate speech is among the "online harms" that are pressing concerns of policymakers, regulators and big tech companies [1]. Reliable real-world hate speech detection models are essential to detect and remove harmful content, and to detect trends and assess the sociological impact of hate speech.

There is an increasing research interest in the automated hate speech detection, as well as competitions and workshops [2]. Hate speech detection is usually modelled as a supervised classification problem, where models are trained to distinguish between examples of hate and normal speech. Most of the current approaches to detect and characterize hate speech focus solely on the content of posts in online social media [3–5]. They do not consider the network structure, nor the roles and types of users generating and retweeting hate speech. A systematic literature review of academic articles on racism and hate speech on social media, from 2014 to

availability: The model for hate speech classification of Slovenian tweets is available at a public language models repository Huggingface at https://huggingface.co/IMSyPP/hate_speech_slo.

**Funding:** The authors acknowledge financial support from the Slovenian Research Agency (research core funding no. P2-103 and P6-0411), the Slovenian Research Agency and the Flemish Research Foundation bilateral research project LiLaH (grant no. ARRS-N6-0099 and FWO-G070619N), and the European Union's Rights, Equality and Citizenship Programme (2014-2020) project IMSyPP (grant no. 875263). The European Commission's support for the production of this publication does not constitute an endorsement of the contents, which reflect the views only of the authors, and the Commission cannot be held responsible for any use which may be made of the information contained therein.

**Competing interests:** The authors have declared that no competing interests exist.

2018 [6], finds that there is a dire need for a broader range of research, going beyond the text-based analyses of overt and blatant racist speech, Twitter, and the content generated mostly in the United States.

In this paper, we go a step further from detecting hate speech from Twitter posts only. We develop and combine a state-of-the-art hate speech classification model with estimates of tweet popularity, retweet communities, influential users, and different types of user accounts (individual, organization, or "problematic"). More specifically, we address the following research questions:

- Are hateful tweets more likely to be retweeted than acceptable tweets?

- Are there meaningful differences between the communities w.r.t. hateful tweets?

- How does the hateful content of a community change over time?

- Which types of Twitter users post more hateful tweets?

As a use case, we demonstrate the results on an exhaustive set of three years of Slovenian Twitter data. We report a number of interesting results which are potentially relevant also for other domains and languages. Hate speech is dominated by offensive tweets, while tweets inciting violence towards target groups are rare. Hateful tweets are retweeted significantly more often than acceptable tweets. There are several politically right-leaning communities which form a super-community. However, about 60% of unacceptable tweets are produced by a single right-leaning community of only moderate size. Institutional and media Twitter accounts post significantly less unacceptable tweets than individual account. Moreover, the main sources of unacceptable tweets are anonymous accounts, and accounts that were closed or suspended during the years 2018–2020.

## Related works

Identifying hate speech and related phenomena in social media has become a very active area of research in natural language processing in recent years. Early work targeted primarily English, and focused on racism and sexism on Twitter [7], harassment in online gaming communities [8], toxicity in Wikipedia talk pages [9], and hate speech and offensive language on Twitter [10]. Results on non-English languages emerged soon after, with early work focusing on, inter alia, hate towards refugees in Germany [11], newspaper comment moderation in Greek [12], Croatian and Slovenian [13], and obscenity and offensiveness of Arabic tweets [14].

There is very little research addressing hate speech in terms of temporal aspects and community structure on Twitter. The most similar research was done on the social media platform Gab (https://Gab.com) [15]. The authors study the diffusion dynamics of the posts by 341,000 hateful and non-hateful users on Gab. The study reveals that the content generated by the hateful users tends to spread faster, farther, and reach a much wider audience as compared to the normal users. The authors also find that hateful users are far more densely connected between themselves, as compared to the non-hateful users. An additional, temporal analysis of hate speech on Gab was performed by taking temporal snapshots [16]. The authors find that the amount of hate speech in Gab is steadily increasing, and that the new users are becoming hateful at an increasingly high rate. Further, the analysis reveals that the hate users are occupying the prominent positions in the Gab network. Also, the language used by the community as a whole correlates better with the language of hateful users than with the non-hateful users.

Our research addresses very similar questions on the Twitter platform. Most of our results on Twitter are aligned with the findings on Gab, however, there are some important differences. Twitter is a mainstream social medium, used by public figures and organizations, while

Gab is an alt-tech social network, with a far-right user base, described as a haven for extremists. We analyse an exhaustive dataset covering all Twitter communication within Slovenia and in the Slovenian language, while Gab covers primarily the U.S. and the English language.

A dynamic network framework to characterize hate communities, focusing on Twitter conversations related to Covid-19, is proposed in [17]. Higher levels of community hate are consistently associated with smaller, more isolated, and highly hierarchical network communities across both the U.S. and the Philippines. In both countries, the average hate scores remain fairly consistent over time. The spread of hate speech around Covid-19 features similar reproduction rates as other Covid-related information on Twitter, with spikes of hate speech at the same times as the highest community-level organization. The identity analysis further reveals that hate in the U.S. initially targets political figures, and then becomes predominantly racially charged. In the Philippines, on the other hand, the targets of hate over time consistently remain political.

In [18], the authors propose a user-centric view of hate speech. They annotate 4,972 Twitter users as hateful or normal, and find that the hateful users differ significantly from the normal users in terms of their activity patterns, word usage, and network structure. In our case, we manually annotate the 890 most influential users for their type, but the level of hate speech of their tweets is automatically assigned by the hate speech classification model.

The relation between political affiliations and profanity use in online communities is reported in [19]. The authors address community differences regarding creation/tolerance of profanity and suggest a contextually nuanced profanity detection system. They report that a political comment is more likely profane and contains an insult or directed insult than a non-political comment.

The work presented here is an extension of our previous research in the area of evolution of retweet communities [20]. The results, obtained on the same Twitter dataset as used here, show that the Slovenian tweetosphere is dominated by politics and ideology [21], that the left-leaning communities are larger, but that the right-leaning communities and users exhibit significantly higher impact. Furthermore, we empirically show that retweet networks change relatively gradually, despite significant external events, such as the emergence of the Covid-19 pandemic and the change of government. In this paper, the detection and evolution of retweet communities is combined with the state-of-the-art models for hate speech classification.

## Structure of the paper

The main results of the paper are in the Structure of the paper Results and discussion section. We first give an overview of the data collected, and how various subsets are used in the Twitter data subsection. In the Hate speech classification subsection we provide a detailed account on training and evaluation of deep learning models. The differences between the detected communities and their roles in posting and spreading hate speech are in the subsection on Communities and hate speech. In subsection Twitter users and hate speech we classify the most influential Twitter users and show the roles of different user types in producing hate speech. In Conclusions we wrap up our combined approach to community evolution and hate speech classification, and present some ideas for future research. The Methods section provides more details regarding the Twitter data acquisition, community detection and evolution, selection of informative timepoints, and retweet influence.

## Results and discussion

This section discusses the main results of the paper. We take two independent approaches of analyzing the same set of Twitter data and then combine them to reveal interesting

conclusions. On one hand, we develop and apply a state-of-the-art machine learning approach to classify hate speech in Twitter posts. On the other hand, we analyze network properties of Twitter users by creating retweet networks, detecting communities, and estimating their influence. This combination allows to distinguish between the communities in terms of how much hate speech they originate and how much they contribute to spreading the hate speech by retweeting. A classification of Twitter users by user types provides additional insights into the structure of the communities and the role of the most influential users in posting and spreading the hate speech.

## Twitter data

Social media, and Twitter in particular, have been widely used to study various social phenomena [22–25]. For this study, we collected a set of almost 13 million Slovenian tweets in the three year period, from January 1, 2018 until December 28, 2020. The set represents an exhaustive collection of Twitter activities in Slovenia. Fig 1 shows the timeline of Twitter volumes and types of speech posted during that period. The hate speech class was determined automatically by our machine learning model. Note a large increase of Twitter activities at the beginning of 2020 when the Covid-19 pandemic emerged, and the left-wing government was replaced by the right-wing government (in March 2020). At the same time, the fraction of hate speech tweets increased.

Our machine learning model classifies Twitter posts into four classes, ordered by the level of hate speech they contain: acceptable, inappropriate, offensive, and violent. It turns out that inappropriate and violent tweets are relatively rare and cannot be reliably classified. Therefore, for this study, all the tweets that are not considered **acceptable** are jointly classified as **unacceptable**. See the next subsection on Hate speech classification for details on the machine learning modelling and extensive evaluations.

Twitter posts are either original tweets or retweets. Table 1 gives a breakdown of the 13-million dataset collected in terms of how different subsets are used in this study. A large subset of

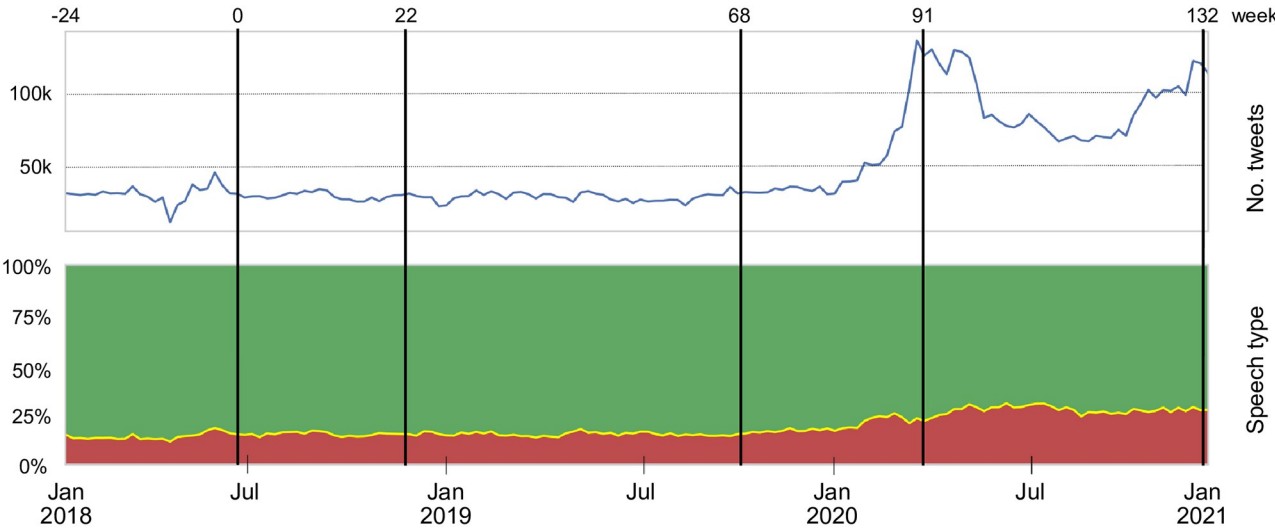

**Fig 1.** Slovenian Twitter posts, collected over the three year period: Weekly volume of collected original tweets (top) and distribution of hate speech classes (bottom). Green area denotes the fraction of acceptable tweets, yellow (barely visible) inappropriate tweets, and red offensive tweets. Tweets inciting violence are not visible due to their low volume (around 0.1%). During the three years, there are 133 time windows from which we create retweet networks. Each window comprises 24 weeks of Twitter data, and subsequent windows are shifted for one week. Vertical lines show five endpoints of automatically selected time windows, with weeks labeled as $t$ = 0, 22, 68, 91, 132.

**Table 1. Slovenian Twitter datasets used in this paper.** Out of almost 13 million tweets collected, a selection of original tweets is used for hate speech annotation, training of classification models, and their evaluation. The retweets are used to create retweet networks, detect communities and influential users.

| Dataset | Period | No. tweets | Role |
|---|---|---|---|
| All tweets | Jan. 2018–Dec. 2020 | 12,961,136 | collection, hate speech classification |
| Original tweets | Jan. 2018–Dec. 2020 | 8,363,271 | hate speech modeling |
| Retweets | Jan. 2018–Dec. 2020 | 4,597,865 | network construction |
| Training set | Dec. 2017–Jan. 2020 | 50,000 | hate speech model learning and cross valid. |
| Evaluation set | Feb. 2020–Aug. 2020 | 10,000 | hate speech model eval. |

the original tweets is used to train and evaluate hate speech classification models. On the other hand, retweets are used to create retweet networks and detect retweet communities. See the subsection on Retweet networks and community detection in Methods for details.

Out of the 13 million tweets, 8.3 million are original tweets, the rest are retweets. Out of 8.3 million original tweets, less than one million are retweeted, and most of them, 7.3 million, are not. Given a hate speech classification model, one can compare two properties of a tweet: is it retweeted or not vs. is it acceptable or unacceptable in terms of hate speech. A proper measure to quantify association between two events is an odds ratio (see Comparing proportions in Methods for definition).

Table 2 provides a contingency table of all the original tweets posted. Less than one million of them were retweeted (12%), possibly several times (therefore the total number of retweets in Table 1 is almost five times larger). On the other hand, the fraction of unacceptable tweets is more than 21%. The odds ratio with 99% confidence interval is 0.678±0.004 and the log odds ratio is -0.388±0.006. This confirms a significant negative correlation between the acceptable tweets and their retweets.

A tweet can be classified solely from the short text it contains. A retweet, on the other hand, exhibits an implicit time dimension, from the time of the original tweet to the time of its retweet. Consequently, retweet networks depend on the span of the time window used to capture the retweet activities. In this study, we use a time span of 24 weeks for our retweet networks, with exponential half-time weight decay of four weeks, and sliding for one week. This results in 133 snapshot windows, labeled with weeks $t = 0, 1, \ldots, 132$ (see Fig 1). It turns out that the differences between the adjacent snapshot communities are small [20]. We therefore implement a heuristic procedure to find a fixed number of intermediate snapshots which maximize the differences between them. For the period of three years, the initial, final, and three intermediate network snapshots are selected, labeled by weeks $t = 0, 22, 68, 91, 132$ (see Fig 1). Details are in subsection Selection of timepoints in Methods.

## Hate speech classification

Hate speech classification is approached as a supervised machine learning problem. Supervised machine learning requires a large set of examples labeled with types of speech (hateful or

**Table 2. Proportions of the (un)acceptable and (not) retweeted tweets over the three year period.** The odds ratio (OR) statistic confirms that acceptable tweets are retweeted significantly less often than the unacceptable tweets.

| Original tweets | Acceptable | Unacceptable | Total (99%) |
|---|---|---|---|
| Retweeted | 708,094 | 270,282 | 978,376 (12%) |
| Not retweeted | 5,866,259 | 1,518,636 | 7,384,895 (88%) |
| Total | 6,574,353 | 1,788,918 | 8,363,271 (99%) |
| | (79%) | (21%) | ln(OR) = -0.388±0.006 |

normal) to cover different textual expressions of speech [26]. Classification models are then trained to distinguish between the examples of hate and normal speech [5]. We pay special attention to properly evaluate the trained models.

The hate speech annotation schema is adopted from the OLID [27] and FRENK [28] projects. The schema distinguishes between four classes of speech on Twitter:

- Acceptable—normal tweets that are not hateful.

- Inappropriate—tweets containing terms that are obscene or vulgar, but they are not directed at any specific person or group.

- Offensive—tweets including offensive generalization, contempt, dehumanization, or indirectly offensive remarks.

- Violent—tweets that threaten, indulge, desire or call for physical violence against a specific person or group. This also includes tweets calling for, denying or glorifying war crimes and crimes against humanity.

The speech classes are ordered by the level of hate they contain, from acceptable (normal) to violent (the most hateful). During the labeling process, and for training the models, all four classes were used. However, in this paper we take a more abstract view and distinguish just between the normal, **acceptable** speech (abbreviated A), and the **unacceptable** speech (U), comprising inappropriate (I), offensive (O) and violent (V) tweets.

We engaged ten well qualified and trained annotators for labeling. They were given the annotation guidelines [29] and there was an initial trial annotation exercise. The annotators already had past experience in a series of hate speech annotation campaigns, including Facebook posts in Slovenian, Croatian, and English. In this campaign, they labeled two sets of the original Slovenian tweets collected: a training and an evaluation dataset.

**Training dataset.**   The training set was sampled from Twitter data collected between December 2017 and January 2020. 50,000 tweets were selected for training different models.

**Out-of-sample evaluation dataset.**   The independent evaluation set was sampled from data collected between February and August 2020. The evaluation set strictly follows the training set in order to prevent data leakage between the two sets and allow for proper model evaluation. 10,000 tweets were randomly selected for the evaluation dataset.

Each tweet was labeled twice: in 90% of the cases by two different annotators and in 10% of the cases by the same annotator. The tweets were uniformly distributed between the annotators. The role of multiple annotations is twofold: to control for the quality and to establish the level of difficulty of the task. Hate speech classification is a non-trivial, subjective task, and even high-quality annotators sometimes disagree on the labelling. We accept the disagreements and do not try to force a unique, consistent ground truth. Instead, we quantify the level of agreement between the annotators (the self- and the inter-annotator agreements), and between the annotators and the models.

There are different measures of agreement, and to get robust estimates, we apply three well-known measures from the fields of inter-rater agreement and machine learning: Krippendorff's Alpha-reliability, accuracy, and F-score.

**Krippendorff's Alpha-reliability** (*Alpha*) [30] was developed to measure the agreement between human annotators, but can also be used to measure the agreement between classification models and a (potentially inconsistent) ground truth. It generalizes several specialized agreement measures, such as Scott's $\pi$, Fleiss' $K$, Spearman's rank correlation coefficient, and Pearson's intraclass correlation coefficient. *Alpha* has the agreement by chance as the baseline, and an instance of it, used here, *ordinal Alpha* takes ordering of classes into account.

**Table 3. The annotator agreement and the overall model performance.** Two measures are used: ordinal Krippendorff's *Alpha* and accuracy (*Acc*). The first line is the self-agreement of individual annotators, and the second line is the inter-annotator agreement between different annotators. The last two lines are the model evaluation results, on the training and the out-of-sample evaluation sets, respectively. Note that the overall model performance is comparable to the inter-annotator agreement.

|  |  | No. of tweets | Overall | |
|---|---|---|---|---|
|  |  |  | *Alpha* | *Acc* |
| Self-agreement |  | 5,981 | 0.79 | 0.88 |
| Inter-annotator agreement |  | 53,831 | 0.60 | 0.79 |
| Classification model | Train.set | 50,000 | 0.61 | 0.80 |
|  | Eval.set | 10,000 | 0.57 | 0.80 |

**Accuracy** (*Acc*) is a common, and the simplest, measure of performance of the model which measures the agreement between the model and the ground truth. Accuracy does not account for the (dis)agreement by chance, nor for the ordering of hate speech classes. Furthermore, it can be deceiving in the case of unbalanced class distribution.

**F-score** ($F_1$) is an instance of the well-known class-specific effectiveness measure in information retrieval [31] and is used in binary classification. In the case of multi-class problems, it can be used to measure the performance of the model to identify individual classes. In terms of the annotator agreement, $F_1(c)$ is the fraction of equally labeled tweets out of all the tweets with label *c*.

Tables 3 and 4 present the annotator self-agreement and the inter-annotator agreement jointly on the training and the evaluation sets, in terms of the three agreement measures. Note that the self-agreement is consistently higher than the inter-annotator agreement, as expected, but is far from perfect.

Several machine learning algorithms are used to train hate speech classification models. First, three traditional algorithms are applied: Naïve Bayes, Logistic regression, and Support Vector Machines with a linear kernel. Second, deep neural networks, based on the Transformer language models, are applied. We use two multi-lingual language models, based on the BERT architecture [32]: the multi-lingual BERT (mBERT), and the Croatian/Slovenian/English BERT (cseBERT [33]). Both language models are pre-trained jointly on several languages but they differ in the number and selection of training languages and corpora.

The training, tuning, and selection of classification models is done by cross validation on the training set. We use blocked 10-fold cross validation for two reasons. First, this method provides realistic estimates of performance on the training set with time-ordered data [34]. Second, by ensuring that both annotations for the same tweet fall into the same fold, we prevent data leakage between the training and testing splits in cross validation. An even more

**Table 4. The annotator agreement and the model performance for individual hate speech classes.** The identification of individual classes is measured by the $F_1$ score. The lines correspond to Table 3. The last three columns give the $F_1$ scores for the three detailed hate speech classes which are merged into a more abstract, Unacceptable class ($F_1(U)$), used throughout the paper. Note relatively low model performance for the Violent class ($F_1(V)$).

|  |  | Acceptable | Unacceptable | Inappropriate | Offensive | Violent |
|---|---|---|---|---|---|---|
|  |  | $F_1(A)$ | $F_1(U)$ | $F_1(I)$ | $F_1(O)$ | $F_1(V)$ |
| Self-agreement |  | 0.92 | 0.87 | 0.62 | 0.85 | 0.69 |
| Inter-annotator agreement |  | 0.85 | 0.75 | 0.48 | 0.71 | 0.62 |
| Classification model | Train.set | 0.85 | 0.77 | 0.52 | 0.73 | 0.25 |
|  | Eval.set | 0.86 | 0.71 | 0.46 | 0.69 | 0.26 |

realistic estimate of performance on yet unseen data is obtained on the out-of-sample evaluation set.

An extensive comparison of different classification models is done following the Bayesian approach to significance testing [35]. Bayesian approach is an alternative to the null hypothesis significance test which has the problem that the claimed statistical significance does not necessarily imply practical significance. One is really interested to answer the following question: What is the probability of the null and the alternative hypothesis, given the observed data? Bayesian hypothesis tests compute the posterior probability of the null and the alternative hypothesis. This allows to detect equivalent classifiers and to claim statistical significance with a practical impact.

In our case, we define that two classifiers are practically equivalent if the absolute difference of their *Alpha* scores is less than 0.01. We consider the results significant if the fraction of the posterior distribution in the region of practical equivalence is less than 5%. The comparison results confirm that deep neural networks significantly outperform the three traditional machine learning models (Naïve Bayes, Logistic regression, and Support Vector Machine). Additionally, language-specific cseBERT significantly outperforms the generic, multi-language mBERT model. Therefore, the cseBERT classification model is used to label all the Slovenian tweets collected in the three year period.

The evaluation results for the best performing classification model, cseBERT, are in Tables 3 and 4. The $F_1$ scores in Table 4 indicate that the acceptable tweets can be classified more reliably than the unacceptable tweets. If we consider classification of the unacceptable tweets in more detail, we can see low $F_1$ scores for the inappropriate tweets, and very low scores for the violent tweets. This low model performance is due to relatively low numbers of the inappropriate (around 1%) and violent tweets (around 0.1%, see Table 5) in the Slovenian Twitter dataset. For this reason, the detailed inappropriate, offensive and violent hate speech classes are merged into the more abstract unacceptable class.

The overall *Alpha* scores in Table 3 show a drop in performance estimate between the training and evaluation set, as expected. However, note that the level of agreement between the best model and the annotators is very close to the inter-annotator agreement. This result is comparable to other related datasets, where the annotation task is subjective and it is unrealistic to expect perfect agreement between the annotators [36, 37]. If one accepts an inherent ambiguity of the hate speech classification task, there is very little room for improvement of the binary classification model.

Table 5 shows the distribution of hate speech classes over the complete Slovenian Twitter dataset. We also provide a breakdown of the unacceptable speech class into its constituent subclasses: inappropriate, offensive, and violent. Offensive tweets are prevailing, inappropriate tweets are rare, and tweets inciting violence are very rare. There is also a considerable difference between the unacceptable original tweets and retweets. Offensive retweets are more frequent (an increase from 20% to 31%), while inappropriate and violent retweets are more rare in comparison to the original tweets.

**Table 5. Distribution of hate speech classes across the original and the retweeted tweets.**

| Tweets | No. of tweets | Acceptable | Unacceptable | | |
|---|---|---|---|---|---|
| | | | Inappropriate | Offensive | Violent |
| Original tweets | 8,363,271 | 6,574,353 (79%) | 88,813 (1.1%) | 1,687,730 (20%) | 12,375 (0.15%) |
| Retweets | 4,597,865 | 3,146,906 (68%) | 20,535 (0.4%) | 1,427,477 (31%) | 2,947 (0.06%) |

## Communities and hate speech

The methods to analyze community evolution through time are described in detail in our related work [20]. They cover formation of retweet networks, community detection, measuring community similarity, selection of coarse-grained timepoints, various measures of influence, and identification of super-communities. In the current paper we use these methods to observe the development of hate speech on Slovenian Twitter during the years 2018–2020.

Fig 2 shows the top seven communities detected at the five selected timepoints. Each node represents a community, where its diameter is a cube-root of the community size (to stifle the large differences in sizes). An edge from the community $C_i$ to $C_j$ indicates average external influence of $C_i$ to $C_j$ in terms of tweets posted by $C_i$ and retweeted by $C_j$. See subsection Retweet influence in Methods for definitions of various types of retweet influence.

The nodes (communities) and edges (external influence links) in Fig 2 form meta-networks. We call communities in meta-networks super-communities. In analogy to a network community, a super-community is a subset of detected communities more densely linked by external influence links than with the communities outside of the super-community. We use this informal definition to identify super-communities in our retweet networks. It is an open research problem, worth addressing in the future, to formalize the definition of super-communities and to design a multi-stage super-community detection algorithm.

In our case, in Fig 2, one can identify three super-communities: the political left-leaning (top), the Sports (middle), and the political right-leaning (bottom) super-community. The prevailing characterization and political orientation of the super-communities is determined by their constituent communities. A community is defined by its members, i.e., a set of Twitter users. A label assigned to a community is just a shorthand to characterize it by its most influential users [20], their types (see subsection Twitter users and hate speech), and tweets they post. Left and Right are generic communities with clear political orientation. SDS is a community

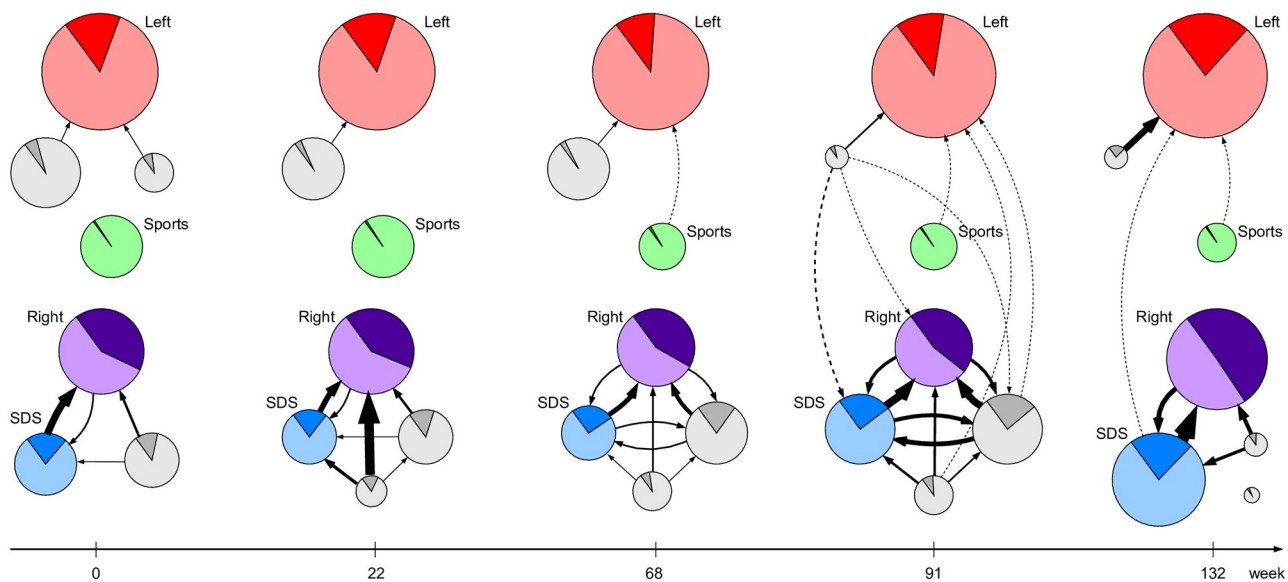

**Fig 2. Fractions of unacceptable tweets posted by different communities.** Nodes are the largest detected communities at timepoints $t$ = 0, 22, . . ., 132. The node size indicates the community size, darker areas correspond to unacceptable tweets, and lighter areas to acceptable tweets. An edge denotes the external influence of community $C_i$ to $C_j$. Linked communities form super-communities: left-leaning (Left, top), Sports (middle), and right-leaning (Right and SDS, bottom).

with a large share of its influential members being politicians and also members of the right-leaning SDS party (Slovenian Democratic Party).

While super-communities exhibit similar political orientation, their constituent communities are considerably different with respect to the hate speech they post. In the following, we compare in detail the largest left-leaning community Left (red), two right-leaning communities, namely Right (violet) and SDS (blue), and a non-political Sports community (green). Left and Right are consistently the largest communities on the opposite sides of the political spectrum. The SDS community was relatively small in the times of the left-leaning governments in Slovenia (until January 2020, $t = 0, 22, 68$), but become prominent after the right-wing government took over (in March 2020, $t = 91, 132$), at the same time as the emergence of the Covid-19 pandemic.

Communities in Fig 2 are assigned proportions of unacceptable tweets they post. Darker areas correspond to fractions of unacceptable tweets, and lighter areas correspond to fractions of acceptable original tweets. Several observations can be made. First, the prevailing Twitter activities are mostly biased towards political and ideological discussions, even during the emergence of the Covid-19 pandemic [21]. There is only one, relatively small, non-political community, Sports. Second, political polarization is increasing with time. Larger communities on the opposite poles grow even larger, and smaller communities are absorbed by them. There are barely any links between the left and right-leaning communities, a characteristics of the echo chambers and political polarization [38]. Third, the fraction of unacceptable tweets posted by the two largest communities, Left and Right, is increasing towards the end of the period. This is clearly visible in Fig 3.

Fig 3 shows the overall increase of unacceptable Twitter posts in the years 2018–2020. Regardless of the community, the fraction of unacceptable tweets in all posts (dotted black line) and in posts that were retweeted (solid black line), are increasing. The same holds for the

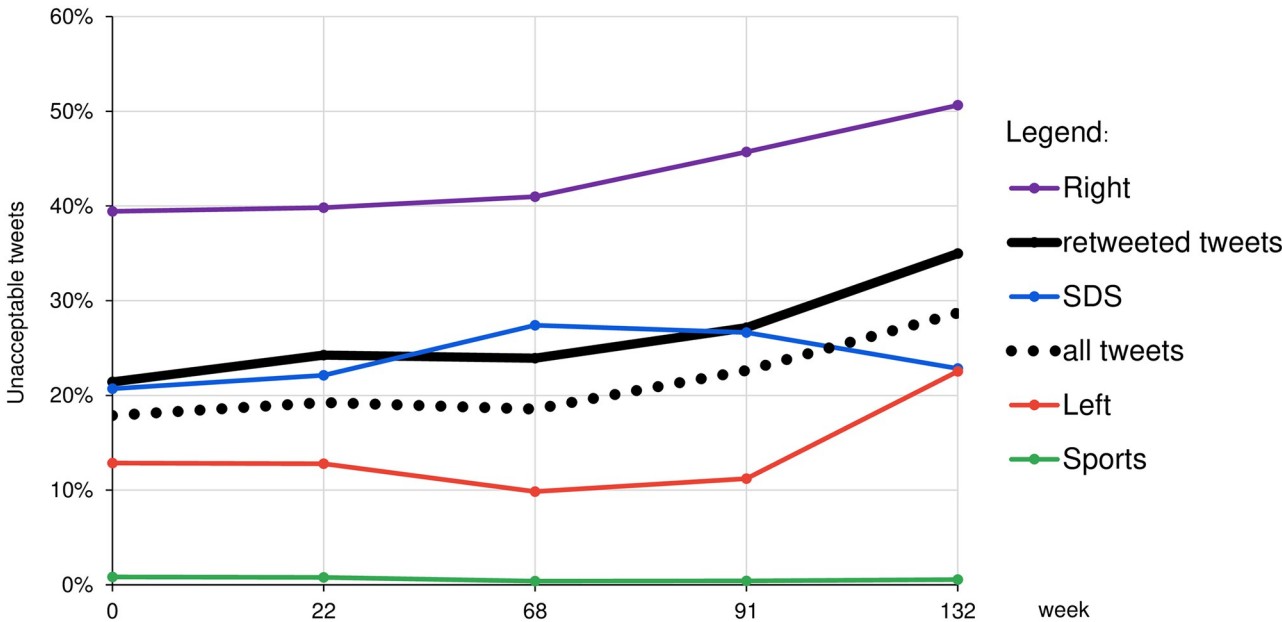

**Fig 3. Fractions of unacceptable tweets posted by the major communities and overall, at weekly timepoints $t = 0, 22, \ldots, 132$.** The solid black line represents all the tweets that were retweeted and are used to form retweet networks and communities. The dotted black line represents all the tweets posted. The largest communities are Left, Right, and SDS. For a comparison, we also show Sports, a small community with almost no unacceptable tweets.

largest Left (red) and Right (violet) communities. However, the right-wing SDS community (blue), shows an interesting change of behaviour. During the left-wing governments in Slovenia, when the SDS party was in opposition (until March 2020, $t = 0, 22, 68$), the fraction of unacceptable tweets they posted was increasing. After SDS became the main right-wing government party (in March 2020, $t = 91, 132$), the fraction of unacceptable tweets they post is decreasing. By the end of 2020 ($t = 132$), SDS and the largest left-leaning community Left converge, both with about 23% of their posted tweets classified as unacceptable. Note that at the same time ($t = 132$), over 50% of the tweets by the Right community is unacceptable. For a comparison, there is also a non-political and non-ideological community Sports (green) with almost no unacceptable tweets.

Fig 4 shows the distribution of unacceptable tweets posted through time. We focus just on the three major communities, Left, Right and SDS. All the remaining communities are shown together as a single Small community (yellow). At any timepoint during the three years, the three major communities post over 80% of all the unacceptable tweets. By far the largest share is due to the Right community, about 60%. The Left and SDS communities are comparable, about 10–20%. However, the three communities are very different in size and in their posting activities.

Fig 5 clearly shows differences between the major communities. We compare the share of unacceptable tweets they post (the leftmost bar), the share of unacceptable tweets they retweet (the second bar from the left), the share of retweet influence (the total number of posted tweets that were retweeted, the third bar from the left), and the size of each community (the rightmost bar). The community shares are computed as the average shares over the five timepoints during the three year period.

The Right community (violet) exhibits disproportional share of unacceptable tweets and retweets w.r.t. its size. Its retweet influence share (the total number of posted tweets that were retweeted) is also larger than its size, which means that its members are more active. However, even w.r.t. to its influence, the share of unacceptable tweets and retweets is disproportional.

The Left community (red) is the most moderate of the three, in terms of unacceptable tweets and retweets. The shares of its posted tweets and retweet influence (weighted out-

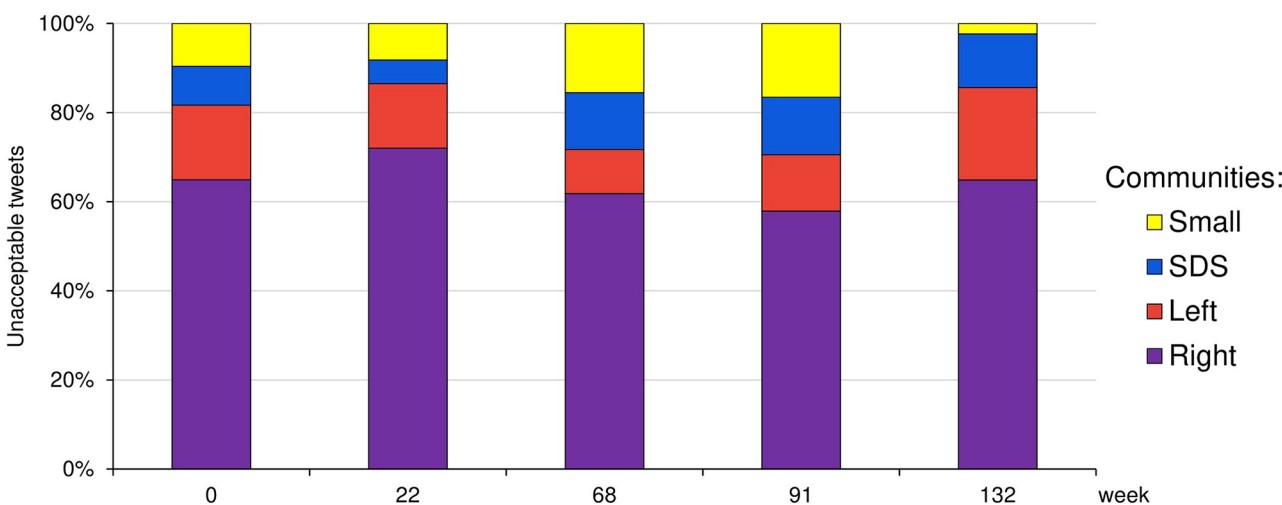

**Fig 4. Distribution of posted unacceptable tweets between the three major communities through time.** Left is the largest, left-leaning community, two right-leaning communities are Right and SDS, and Small denotes all the remaining smaller communities. Weekly timepoints are marked by $t = 0$, 22, . . ., 132. The Right community posts the largest share of the unacceptable tweets, over 60% at four out of five timepoints. The Left and SDS communities are comparable, each with the share of about 10–20% of all unacceptable tweets posted.

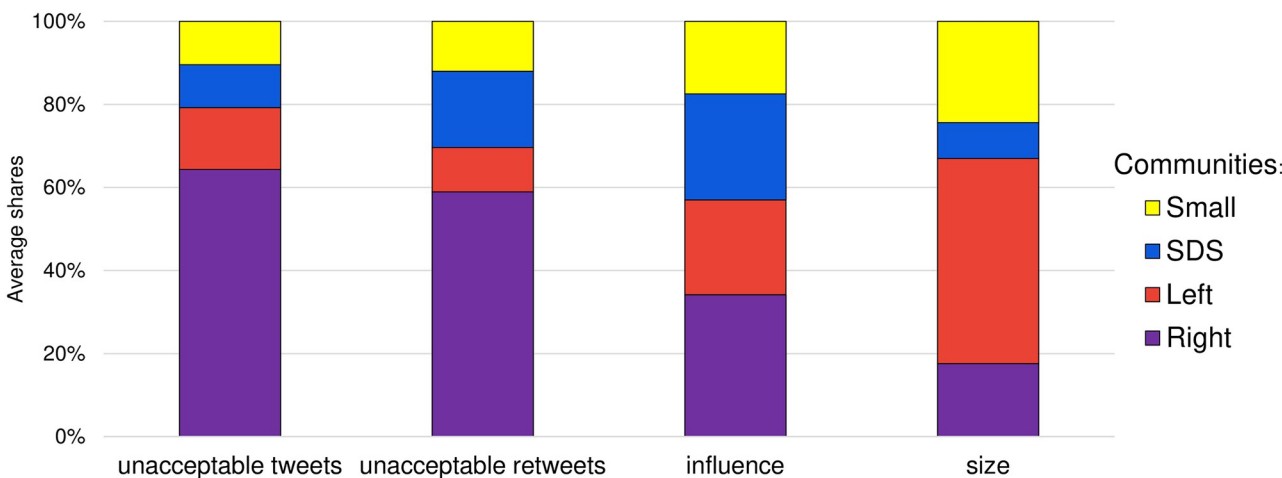

**Fig 5. Comparison of the three major communities in terms of four different properties.** Each bar is composed of the Right, Left, SDS, and the remaining Small communities, from bottom to top. Bars correspond to the average shares (over the five weekly timepoints) of posted unacceptable tweets, retweeted unacceptable tweets, community influence (weighted out-degree), and size of the community, from left-to-right, respectively.

degree) w.r.t. its size, are lower in comparison to the Right and SDS communities. This indicates that its members are, on average, less active and less influential.

The SDS community (blue) posts about the same share of unacceptable tweets as is expected for its size. However, its share of unacceptable retweets is larger. It is also very active and the most influential of the three, and in this respect its share of unacceptable tweets posted is lower w.r.t. its influence share.

The differences between proportions of various community aspects can be quantified by Cohen's $h$ [39]. Cohen's $h$ quantifies the size of the difference, allowing one to decide if the difference is meaningful. Namely, the difference can be statistically significant, but too small to be meaningful. See subsection Comparing proportions in Methods for details. Table 6 gives the computed $h$ values for the three major communities. The results are consistent with our interpretations of Fig 5 above.

## Twitter users and hate speech

The analysis in the previous subsection points to the main sources of unacceptable tweets posted and retweeted at the community level. In this subsection, we shed some light on the composition of the major communities in terms of the user types and their individual influence.

We estimate a Twitter user influence by the retweet h-index [40], an adaptation of the well known Hirsch index [41] to Twitter. A user with a retweet index $h$ posted $h$ tweets and each of

**Table 6. Comparison of the three major communities by Cohen's h.** The headings denote the first property (the proportion $p_1$) vs. the second property (the proportion $p_2$). The values of $h$ in the body have the following interpretation: positive sign of $h$ shows that $p_1 > p_2$, and the value of $|h|$ indicates the effect size. In bold are the values of $h > 0.50$, indicating at least medium effect size.

| Community | Unacc. tweets vs. Size | Unacc. retweets vs. Size | Influence vs. Size | Unacc. tweets vs. Influence |
|---|---|---|---|---|
| Right | **1.00** | **0.88** | 0.38 | **0.61** |
| Left | -0.77 | -0.89 | -0.56 | -0.20 |
| SDS | 0.06 | 0.29 | 0.46 | -0.41 |

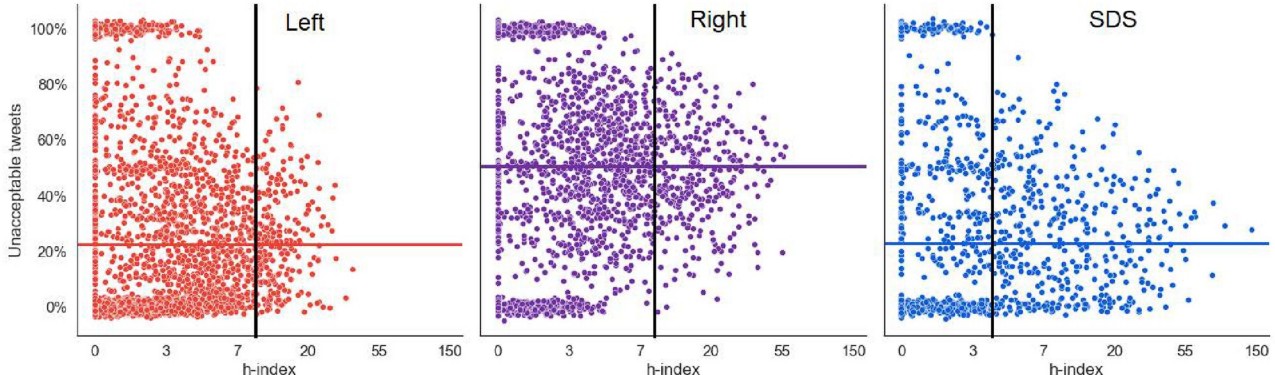

**Fig 6. Scatter plot of the three major communities at the last timepoint, *t* = 132.** Each point represents a Twitter user, with its retweet h-index and a fraction of unacceptable tweets posted. Horizontal lines show the average fraction of unacceptable tweets per community. Vertical bars delimit the low influence from the high influence Twitter users. For the most influential users, with an h-index right of the vertical bar, the user types are determined (see Table 8).

them was retweeted at least *h* times. See subsection Retweet influence in Methods for details. It was already shown that members of the right-leaning super-community exhibit much higher h-index influence than the left-leaning users [20]. Also, influential users rarely switch communities, and when they do, they stay within their left- or right-leaning super-community.

In Fig 6 we show the distribution of Twitter users from the three major communities detected at the end of the three year period (*t* = 132). The scatter plots display individual users in terms of their retweet h-index (x-axis, logarithmic scale) and fraction of unacceptable tweets they post (y-axis). The average proportions of unacceptable tweets posted by the community members are displayed by horizontal lines. The results are consistent with Fig 3 at the last timepoint *t* = 132, where the Left, Right and SDS communities post 23%, 51% and 23% of unacceptable tweets, respectively. More influential users are at the right hand-side of the plots. Consistent with Fig 5, the members of the SDS and Right communities are considerably more influential than the members of the Left. In all the communities, there are clusters of users which post only unacceptable tweets (at the top), or only acceptable tweets (at the bottom). However, they are not very prolific nor do they have much impact, since their retweet h-index is very low. Vertical bars delimit the low influence from the high influence Twitter users.

Fig 6 shows that the distribution of influence in terms of retweet h-index is different between the three communities. We compute the concentration of influence by the Gini coefficient, a well-known measure of income inequality in economics [42]. Gini coefficient of 0 indicates perfectly equal distribution of influence, and Gini of 1 indicates the extreme, i.e., all the influence in concentrated in a single user. The results in Table 7 show that the highest concentration of influence is in the SDS community, followed by the Right, and that the Left community has more evenly distributed influence.

For the most influential Twitter users, right of the vertical bars in Fig 6, we inspect their type and their prevailing community during the whole time period. They are classified into

**Table 7. Gini coefficients of influence distribution for the three major communities at the last timepoint, *t* = 132.**

| Community | Gini coefficient |
|---|---|
| Left | 0.50 |
| Right | 0.57 |
| SDS | 0.64 |

**Table 8. Twitter user types and their prevailing communities.** The top 890 users from the major communities, ranked by the retweet h-index, are classified into different types. When possible (in over 72% of the cases) the prevailing community across the five timepoints is determined. The rest of the users shift between different communities through time. There is an interesting transition community Left→SDS that corresponds to the government transition from the left-wing to the right-wing, and consists mostly of the governmental institutions.

| User type subtype | Share | Prevailing community | | | |
|---|---|---|---|---|---|
| | | **Left** | **Right** | **SDS** | **Left→SDS** |
| Individual | 486 (55%) | 137 (59%) | 85 (38%) | 104 (60%) | 3 (20%) |
| Politician | 143 | 20 | 16 | 72 | 3 |
| Public figure | 101 | 40 | 22 | 8 | 0 |
| Journalist | 100 | 41 | 12 | 11 | 0 |
| Other | 142 | 36 | 35 | 13 | 0 |
| Organization | 129 (14%) | 41 (18%) | 9 0(4%) | 36 (21%) | 12 (80%) |
| Institution | 59 | 20 | 3 | 9 | 10 |
| Media | 46 | 16 | 4 | 15 | 1 |
| Political party | 24 | 5 | 2 | 12 | 1 |
| Unverified | 275 (31%) | 55 (23%) | 130 (58%) | 32 (19%) | 0 0(0%) |
| Anonymous | 148 | 37 | 47 | 16 | 0 |
| Closed | 95 | 14 | 58 | 13 | 0 |
| Suspended | 32 | 4 | 25 | 3 | 0 |
| Total | 890 (99%) | 233 (99%) | 224 (99%) | 172 (99%) | 15 (99%) |

three major categories: Individual, Organization, and Unverified. The Unverified label is not meant as the opposite of the Twitter verification label, but just lumps together the users for which the identity was unclear (Anonymous), their accounts were closed (Closed) or suspended by Twitter (Suspended). The Individual and Organization accounts are further categorized into subtypes.

Table 8 provides the categorization of 890 users into types and subtypes. We selected the top users from the major communities, ranked by their retweet h-index. When the user did not switch between the communities (in 644 out of 890 cases, 72%) we assign its prevailing community across the whole time period from the community membership at individual timepoints. We introduce an additional transition community, Left→SDS, that encompasses Twitter accounts which switched from the Left community to the current government SDS community at the time of the government transition from the left-wing to the right-wing. This transition community consists mostly of governmental accounts (ministries, army, police, etc.) and demonstrates surprisingly well how detected communities in time reflect the actual changes in the political landscape.

The 890 users, classified into different types, represent less than 5% of all the users active on Slovenian Twitter during the three year period. However, they exhibit the bulk of the retweet influence. They posted almost 10% of all the original tweets, and, even more indicative, over 50% of all the retweeted tweets were authored by them. The details are given in Table 9.

Fig 7 shows how many unacceptable tweets are posted by different user types and subtypes. Over 40% of tweets posted by unverified accounts are unacceptable. In this category, the suspended accounts lead with over 50% of the tweets classified as unacceptable. This demonstrates

**Table 9. Influential users.** The share of influential users in terms of their number, the original tweets they post, and their tweets that were retweeted.

| | Users | | Original tweets | | Retweeted tweets | |
|---|---|---|---|---|---|---|
| All | 18,821 | | 8,363,271 | | 978,376 | |
| Influential | 890 | (4.7%) | 812,862 | (9.7%) | 529,110 | (54.1%) |

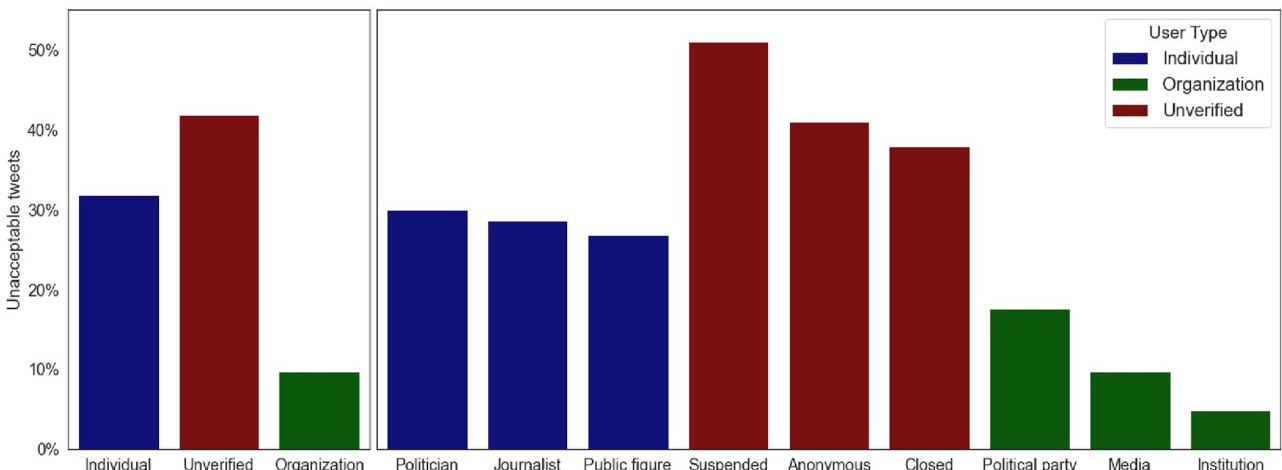

**Fig 7. Fractions of unacceptable tweets posted by different types of users.** The left bar chart shows major user types. Each major user type consists of three subtypes, shown at the right bar chart with the same color. Individual bars indicate the fraction of all tweets posted by the user (sub)type that are unacceptable.

that Twitter is doing a reasonable job at suspending problematic accounts, and that our hate speech classification model is consistent with the Twitter criteria. Note that on the global level, Twitter is suspending an increasing number of accounts (776,000 accounts suspended in the second half of 2018, and one million accounts suspended in the second half of 2020).

Individual accounts, where politicians dominate between the influential users, post over 30% of tweets as unacceptable. Organizational accounts post mostly acceptable tweets (90%). In this category, accounts from the political parties dominate, with a fraction of 17% of their tweets being unacceptable. There is an interesting difference between the individual journalists and media organizations. Official media accounts post about 10% of tweets as unacceptable, while for the influential journalists, this fraction is 28%.

At the end, we can provide a link between the major retweet communities and the user types. We use Cohen's $h$ again to quantify the differences between the representation of the main user types in the communities and their overall share. The community-specific proportions (first property) and the overall share (second property) of the main user types are taken from Table 8. The $h$ values in Table 10 then quantify which user types post disproportional fractions of unacceptable tweets (first column in Table 10), and in which communities are they disproportionately represented (columns 2-4 in Table 10).

Results in Table 10 confirm that the Unverified accounts produce a disproportionate fraction of unacceptable tweets, and that they are considerably over-represented in the Right community. On the other hand, Individual and Organization accounts are under-represented in the Right community.

**Table 10. Comparison of the three major user types by Cohen's $h$.** The headings denote the first property (the proportion $p_1$) vs. the second property (the proportion $p_2$). The values of $h$ in the body have the following interpretation: positive sign of $h$ shows that $p_1 > p_2$, and the value of $|h|$ indicates the effect size. In bold is the value of $h > 0.50$, indicating at least medium effect size.

| User type | Unacc. tweets vs. Share | Left vs. Share | Right vs. Share | SDS vs. Share |
|---|---|---|---|---|
| Individual | -0.04 | 0.08 | -0.34 | 0.12 |
| Organization | -0.26 | 0.08 | -0.38 | 0.17 |
| Unverified | 0.21 | -0.16 | **0.55** | -0.29 |

## Conclusions

Retweets play an important role in revealing social ties between the Twitter users. They allow for the detection of communities of like-minded users and super-communities linked by the retweet influence links. In our Slovenian Twitter dataset, the two main super-communities show clear political polarization between the left and right-leaning communities [20]. The right-leaning communities are closely linked, and exhibit significantly higher retweet influence than the left-leaning communities. This is consistent with the findings about the European Parliament [43] and polarization during the Brexit referendum [40]. However, in terms of hate speech, the super-communities are not homogeneous, and there are large differences between the communities themselves.

Regarding the hate speech classification, we demonstrate that the best model reaches the inter-annotator agreement. This means that a model with such level of performance can replace a human annotator, and that without additional information, the model cannot be improved much. The additional information, if properly taken into account, might be in the form of a context. Textual context, such as previous tweets or a thread, is difficult to incorporate in the machine learning models. The user context, on the other hand, can provide additional features about the user history and community membership, and seems very relevant and promising for better hate speech classification.

Our hate speech classification model distinguishes between three classes of hate speech on Twitter: inappropriate, offensive, and violent. Specially tweets inciting violence, and directed towards specific target groups, are essential to detect since they may be subject to legal actions. However, in our training data sample of 50,000 tweets, the annotators found only a few 100 cases of violent tweets. The evaluation results show that the model cannot reliably detect violent hate speech, therefore we classified all three classes of hate speech together, as unacceptable tweets. Our previous experience in learning Twitter sentiment models for several languages [36] shows that one needs several 1,000 labelled tweets to construct models which approach the quality of human annotators. This calls for additional sampling of a considerably larger set of potentially violent tweets, which should be properly annotated and then used for model training.

Another dimension of hate speech analysis are the topics which are discussed. The results of topic detection on Slovenian Twitter show that political and ideological discussions are prevailing, accounting for almost 45% of all the tweets [21]. The sports-related topic, for example, is subject of only about 12% of all the tweets, and 90% of them are acceptable. This is also consistent with the very low fraction of unacceptable tweets posted by the Sports community in Fig 3. The distribution of topics within the detected communities, the levels of topic-related hate speech, and the evolution through time are some of the interesting results reported in [21].

We identify one, right-leaning, community of moderate size which is responsible for over 60% of unacceptable tweets. In addition, we show that this community consists of a disproportional share of anonymous, suspended, or already closed Twitter accounts which are the main source of hate speech. The other right-leaning community, corresponding to the main party of the current right-wing Slovenian government, shows more moderation, in particular after it took over from the left-wing government in March 2020. While these results are specific for the Slovenian tweetosphere, there are two lessons important for other domains and languages. One is the concept of super-communities which can be identified after the standard community detection process [20, 44], and share several common properties of the constituent communities. Another is the insight that hate speech is not always evenly spread within a super-community, and that it is important to analyze individual communities and different types of users.

## Methods

### Data collection

The three years of comprehensive Slovenian Twitter data cover the period from January 1, 2018 until December 28, 2020. In total, 12,961,136 tweets were collected, indirectly through the public Twitter API. The data collection and data sharing complies with the terms and conditions of Twitter. We used the TweetCaT tool [45] for Twitter data acquisition.

The TweetCaT tool is specialized on harvesting Twitter data of less frequent languages. It searches continuously for new users that post tweets in the language of interest by querying the Twitter Search API for the most frequent and unique words in that language. Once a set of new potential users posting in the language of interest are identified, their full timeline is retrieved and the language identification is run over their timeline. If it is evident that specific users post predominantly in the language of interest, they are added to the user list and their tweets are being collected for the remainder of the collection period. In the case of Slovenian Twitter, the collection procedure started in August 2017 and is still running. As a consequence, we are confident that the full Slovenian tweetosphere is covered in the period of this analysis.

### Comparing proportions

Odds ratio and Cohen's *h* are two measures of association and effect size of two events. Odds ratio can be used when both events are characterized by jointly exhaustive and mutually exclusive partitioning of the sample. Cohen's *h* is used to compare two independent proportions.

**Odds ratio.** An odds ratio is a statistic that quantifies the strength of the association between two events. The (natural logarithm of the) odds ratio *L* of a sample, and its approximate standard error *SE* are defined as:

$$L = ln\left(\frac{n_{11} \cdot n_{00}}{n_{10} \cdot n_{01}}\right), \quad SE = \sqrt{\frac{1}{n_{11}} + \frac{1}{n_{00}} + \frac{1}{n_{10}} + \frac{1}{n_{01}}},$$

where $n_{ij}$ are the elements of a $2 \times 2$ contingency table. A non-zero log odds ratio indicates correlation between the two events, and the standard error is used to determine its significance.

**Cohen's *h*.** The difference between two independent proportions (probabilities) can be quantified by Cohen's *h* [39]. For two proportions, $p_1$ and $p_2$, Cohen's *h* is defined as the difference between their "arcsine transformations":

$$h = 2\arcsin\sqrt{p_1} - 2\arcsin\sqrt{p_2}.$$

The sign of *h* shows which proportion is greater, and the magnitude indicates the effect size. Cohen [39, p. 184–185] provides the following rule of thumb interpretation of *h*: 0.20– small effect size, 0.50–medium effect size, and 0.80–large effect size.

### Retweet networks and community detection

Twitter provides different forms of interactions between the users: follows, mentions, replies, and retweets. The most useful indicator of social ties between the Twitter users are retweets [44, 46]. When a user retweets a tweet, it is distributed to all of its followers, just as if it were an originally authored tweet. Users retweet content that they find interesting or agreeable.

A retweet network is a directed graph. The nodes are Twitter users and edges are retweet links between the users. An edge is directed from the user *A* who posts a tweet to the user *B* who retweets it. The edge weight is the number of retweets posted by *A* and retweeted by *B*.

For the whole three year period of Slovenian tweets, there are in total 18,821 users (nodes) and 4,597,865 retweets (sum of all weighted edges).

To study dynamics of the retweet networks, we form several network snapshots from our Twitter data. In particular, we select a network observation window of 24 weeks (about six months), with a sliding window of one week. This provides a relatively high temporal resolution between subsequent networks, but in the next subsection Selection of timepoints we show how to select the most relevant intermediate timepoints. Additionally, in order to eliminate the effects of the trailing end of a moving network snapshot, we employ an exponential edge weight decay, with half-time of 4 weeks.

The set of network snapshots thus consists of 133 overlapping observation windows, with temporal delay of one week. The snapshots start with a network at $t = 0$ (January 1, 2018–June 18, 2018) and end with a network at $t = 132$ (July 13, 2020–December 28, 2020) (see Fig 1).

Informally, a network community is a subset of nodes more densely linked between themselves than with the nodes outside the community. A standard community detection method is the Louvain algorithm [47]. Louvain finds a partitioning of the network into communities, such that the modularity of the partition is maximized. However, there are several problems with the modularity maximization and stability of the Louvain results [48]. We address the instability of Louvain by applying the **Ensemble Louvain** algorithm [20, 49]. We run 100 trials of Louvain and compose communities with nodes that co-occur in the same community above a given threshold, 90% of the trials in our case. This results in relatively stable communities of approximately the same size as produced by individual Louvain trials. We run the Ensemble Louvain on all the 133 undirected network snapshots, resulting in 133 network partitions, each with slightly different communities.

## Selection of timepoints

There are several measures to evaluate and compare network communities. We use the BCubed measure, extensively evaluated in the context of clustering [50]. BCubed decomposes evaluation into calculation of precision and recall of each node in the network. The precision (*Pre*) and recall (*Rec*) are then combined into the $F_1$ score, the harmonic mean:

$$F_1 = 2 \, \frac{Pre \cdot Rec}{Pre + Rec}.$$

Details of computing *Pre* and *Rec* for individual nodes, communities and network partitions are in [20]. We write $F_1(P_i|P_j)$ to denote the $F_1$ difference between the partitions $P_i$ and $P_j$. The paper also provides a sample comparison of BCubed with the Adjusted Rand Index (ARI) [51] and Normalized Mutual Information (NMI) [52]. Our $F_1$ score extends the original BCubed measure to also account for new and disappearing nodes, and is different and more general than the $F_1$ score proposed by Rossetti [53].

The weekly differences between the network partitions are relatively small. The retweet network communities do not change drastically at this relatively high time resolution. Moving to lower time resolution means choosing timepoints which are further apart, and where the network communities exhibit more pronounced differences.

We formulate the timepoint selection task as follows. Let us assume that the initial and final timepoints are fixed (at $t = 0$ and $t = n$), with the corresponding partitions $P_0$ and $P_n$, respectively. For a given $k$, select $k$ intermediate timepoints such that the differences between the corresponding partitions are maximized. The number of possible selections grows exponentially with $k$. Therefore, we implement a simple heuristic algorithm which finds the $k$ (non-optimal) timepoints. The algorithm works top-down and starts with the full, high resolution timeline

with $n + 1$ timepoints, $t = 0, 1, \ldots, n$ and corresponding partitions $P_t$. At each step, it finds a triplet of adjacent partitions $P_{t-1}, P_t, P_{t+1}$ with minimal differences (i.e., maximum $F_1$ scores):

$$max(F_1(P_t|P_{t-1}) + F_1(P_{t+1}|P_t)).$$

The partition $P_t$ is then eliminated from the timeline:

$$P_0, \ldots, P_{t-1}, P_t, P_{t+1}, \ldots, P_n \ \mapsto \ P_0, \ldots, P_{t-1}, P_{t+1}, \ldots, P_n.$$

At the next step, the difference $F_1(P_{t+1}|P_{t-1})$ fills the gap of the eliminated timepoint $P_t$. The step is repeated until there are $k$ (non-optimal) intermediate timepoints. The heuristic algorithm thus requires $n - 1 - k$ steps.

For our retweet networks, we fix $k = 3$, which provides much lower, but still meaningful time resolution. This choice results in a selection of five network partitions $P_t$ at timepoints $t = 0, 22, 68, 91, 132$.

## Retweet influence

Twitter users differ in how prolific they are in posting tweets, and in the impact these tweets make on the other users. One way to estimate the influence of Twitter users is to consider how often their tweets are retweeted. Similarly, the influence of a community can be estimated by the total number of retweets of tweets posted by its members. Retweets within the community indicate **internal influence**, and retweets outside of the community indicate **external influence** [44, 54].

Let $W_{ij}$ denote the sum of all weighted edges between communities $C_i$ and $C_j$. The average community influence $I$ is defined as:

$$I(C_i) = \frac{\sum_j W_{ij}}{|C_i|},$$

i.e., the weighted out-degree of $C_i$, normalized by its size. The influence $I$ consists of the internal $I_{int}$ and external $I_{ext}$ component, $I = I_{int} + I_{ext}$, where

$$I_{int}(C_i) = \frac{W_{ii}}{|C_i|},$$

and

$$I_{ext}(C_i, C_j) = \frac{\sum_{i \neq j} W_{ij}}{|C_i|}.$$

We compute internal and external influence of the retweet communities detected at the selected timepoints $t = 0, 22, 68, 91, 132$. Fig 2 shows the communities and the external influence links between the detected communities. One can observe a formation of super-communities, with closely linked communities. There are two super-communities, the political left-leaning and right-leaning, and an apolitical Sports.

Weighted out-degree is a useful measure of influence for communities. For individual Twitter users, a more sophisticated measure of influence is used. The user influence is estimated by their **retweet h-index** [40, 55], an adaptation of the well known Hirsch index [41] to Twitter. The retweet h-index takes into account the number of tweets posted, as well as the impact of individual tweets in terms of retweets. A user with an index of $h$ has posted $h$ tweets and each of them was retweeted at least $h$ times.

## Author Contributions

**Conceptualization:** Igor Mozetič.

**Data curation:** Nikola Ljubešić.

**Software:** Bojan Evkoski, Andraž Pelicon.

**Supervision:** Petra Kralj Novak.

**Validation:** Andraž Pelicon.

**Visualization:** Bojan Evkoski.

**Writing – original draft:** Igor Mozetič.

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
