## [Decision Letter · Decision Letter 0]

16 Jul 2021

PONE-D-21-16242

Retweet communities reveal the main sources of hate speech

PLOS ONE

Dear Dr. Mozetič,

Thank you for submitting your manuscript to PLOS ONE. After careful consideration, we feel that it has merit but does not fully meet PLOS ONE’s publication criteria as it currently stands. Therefore, we invite you to submit a revised version of the manuscript that addresses the points raised during the review process.

We look forward to receiving your revised manuscript.

Kind regards,

Antonio Scala, PhD

Academic Editor

PLOS ONE

Journal Requirements:

2. Please include in your Methods section a statement on whether the data collection and data sharing complies with the terms and conditions of Twitter.

"The authors acknowledge financial support from the Slovenian Research Agency 533

(research core funding no. P2-103 and P6-0411), from the European Union's Rights, 534

Equality and Citizenship Programme (2014-2020) project IMSyPP (Innovative 535

Monitoring Systems and Prevention Policies of Online Hate Speech, grant no. 875263), 536

and from the Slovenian Research Agency and the Flemish Research Foundation bilateral 537

research project LiLaH (The linguistic landscape of hate speech on social media, grant 538

no. ARRS-N6-0099 and FWO-G070619N). 539"

5. We note that your manuscript contains information which identifies and names individual Twitter users. As per the PLOS ONE policy (http://journals.plos.org/plosone/s/submission-guidelines#loc-human-subjects-research) on papers that include identifying, or potentially identifying, information, the individual(s) must be informed of the terms of the PLOS open-access (CC-BY) license and provide specific permission for publication of these details under the terms of this license. Please download the Consent Form for Publication in a PLOS Journal (http://journals.plos.org/plosone/s/file?id=8ce6/plos-consent-form-english.pdf). The signed consent form should not be submitted with the manuscript, but should be securely filed in the individual's case notes. Please amend the methods section and ethics statement of the manuscript to explicitly state that the users have provided consent for publication: “The individuals named in this manuscript have given written informed consent (as

outlined in PLOS consent form) to publish these case details”.

Alternatively, if it is not possible to obtain these permissions, we ask that you modify your manuscript to remove any information that could identify individual users.

Reviewers' comments:

Reviewer's Responses to Questions

**Comments to the Author**

1. Is the manuscript technically sound, and do the data support the conclusions?

Reviewer #1: Yes

Reviewer #2: Yes

2. Has the statistical analysis been performed appropriately and rigorously? 

Reviewer #1: Yes

Reviewer #2: Yes

3. Have the authors made all data underlying the findings in their manuscript fully available?

Reviewer #1: Yes

Reviewer #2: Yes

4. Is the manuscript presented in an intelligible fashion and written in standard English?

Reviewer #1: Yes

Reviewer #2: Yes

5. Review Comments to the Author

Reviewer #1: # Retweet Communities reveal the main sources of hate speech

**Overview:** The following work addresses the problem of identifying the main sources of hate speech on Twitter. To accomplish this, the authors employed a dataset concerning the Slovenian Twitter debate on a three-year period (2018-2020), with a total of 12 961 136 pieces of contents. The experiments proposed provided results on two aspects of this topic. First, they propose an automatic way of annotating tweets in relationship with their textual content, using recent results from the BERT architecture. Then, they moved the attention on the community and user aspects posting and retweeting the previously labeled tweets, exploring a possible relationship between hateful tweets, the political leaning of communities, and their evolution over time.

**Principal claims and significance for the discipline:**

- In terms of hate speech detection, the model the authors propose was able to classify unacceptable tweets from their counterparts with promising results comparable with the ones produced by human annotators.

- In terms of community detection, the authors mapped the evolution of interconnection of users and their aggregation, resulting in the presence of three main super communities mainly related with the Left and Right-leaning, plus a smaller one concerning sports topics. In relationship with the amount and the type of information proposed from those communities, the authors found that the Right super community is linked to a greater spreading of unacceptable tweets despite their smaller size. The right-leaning side shows a greater ability to influence users, compared with its Left counterpart. Moreover, a small right-leaning community seems responsible for 60% of the unacceptable tweets proposed by the Right super community. The authors denote how these results, especially the concept of super-communities and the fact that hate speech is not necessarily shared within a super-community, are generally applicable in contexts that are not related to the Slovenian debate.

**Literature positioning:** the findings proposed are properly placed in the context of the previous literature and they are consistent with the research line of the authors, which can be inferred by looking at the citations of the authors' works.

**Data availability:** the authors, in line 372, claim that the dataset is available at the following link: https://www.clarin.si/repository/xmlui/handle/11356/1423. However, it looks like the repository is not publicly available, even if logged in through an institution. From a Google search about Slovenian Twitter dataset, I find the following dataset which seems employed in the study. The dataset I found is available at this link: https://www.clarin.si/repository/xmlui/handle/11356/1398

**Reproducibility:** the reproducibility of the paper is not generally guaranteed, mainly for the absence of a public access to the dataset. In terms of the explanation of the methodology, the paper requires minor corrections to this part.

**Structural observations:**

The manuscript is well organized in its structure and it is compliant with the structure of the journal. However, some corrections needs to be made:

1. Introduction - it seems to lack some brief introduction about the hate speech context, such as the problem it arises, the consequences, and similar. I would spend some lines at the beginning to briefly give some context to this topic to the reader.

2. line 2 and 3: I find the first clause a strong statement, especially if we consider where it is placed and the lack of literature in support. I encourage you to rephrase it and/or find works that can sustain this opening.

3. Twitter data, line 111 and 112: "various social phenomena" could be less vague by citing some influential works that provide an example of the aforementioned phenomena. I suggest to cite works like this one: https://journals.plos.org/plosone/article?id=10.1371/journal.pone.0234689

4. Results and discussion, Table 1: "Dec. 2017" is ambiguous. In the data collection section, it is mentioned that TweetCat started to collect Slovenian tweets at the end of 2017 but, at the same time, the Twitter data covered the period from 2018-2020. Therefore, would spend some words to clarify the presence of this "Dec. 2017" term.

5. line 131 and 132: "Out of [...] not". I suggest rephrasing it to be less qualitative.

6. Conclusions, line 325: the part about hate speech detection should have been placed before the community one in order to maintain the same order with the Results section

7. Methods: in Cohen's h, I would move the term "(probabilities)" to the first occurrence of proportions in the caption.

8. line 382: "Cohen provides [..] interpretation", I think it would be more sound to cite again the paper providing this rule of thumb.

**Methodological observations:**

- Twitter data, line 115: "levels of hate speech" refers to a result of the ML part. At this point, it should be pointed out how those levels were measured.

- Communities and hate speech, line 163: "tightly linked" is too vague. Which measures or analysis led to the definition of "tightly linked"? A better explanation is required, especially in this part where the creation of super-communities is crucial for the results presented below in the paper.

- Twitter users and hate speech, line 250: the clause "the results are consistent with Figure 3" requires to be rephrased in order to provide more evidence about this consistency.

- Line 267: "A prevailing community [...] is also determined". How?

- Hate speech classification, line 386-387: "Supervised machine learning requires a large set of examples labeled for hate speech" is ambiguous. Probably, you are only referring to the context of hate speech. However, this sentence should be motivated by citing works that sustain this "rule of thumb".

- Methods - The heuristic algorithm proposed to find the k timepoints requires better explanation and editing. It could be provided in a pseudocode way and/or by providing a visualization of a general iteration

**Plot observations:**

- Figure 1: axis labels are required.

- Figure 3: y-axis label and legend title are required.

- Figure 4: y-axis and legend title are required. Furthermore, its caption should be modified in order to be less colloquial and more informative.

- Figure 5: I am not convinced by this plot. It is too qualitative and compares different metrics. Cohen's h gives an answer on this side, but it requires checking the table containing the results, meaning that the figure itself does not provide precise information. It is visually hard to understand the differences between communities on the metrics proposed if a scale is missing. Therefore, the explanation proposed from lines 212 to 231 may be not so clear to the reader as the paper states. My proposal is to switch to a figure containing a bar plot for each metric of the different measurement units. In the end, a title for the legend is missing.

- Figure 6: I would add the explanation of the vertical line in the caption.

- Figure 7: The figure lacks a legend and an x-axis label. The caption should be more descriptive, briefly describing the results and being less colloquial with the definition of the "refined subtypes".

**Linguistic observations:**

1. Abstract: the usage of "however" in the last sentence is ambiguous. The sentence in which this word is used does not present any contradiction with respect to the previous one but, in some way, it enforces the statement. Therefore, I would suggest changing to "Indeed", "In fact" or similar.

2. Related work, line 29: change to Related works.

3. line 33: I find the term "Work" colloquial in the context of a paper. I suggest changing to "Results" since we rely on the outcomes of the works.

4. line 44: since only one further work is cited, I would suggest modifying "Another" to "An additional", becoming less colloquial at the same time.

5. Results and discussion, line 104: "The combination then" should be changed to "This combination"

6. Table 1: I would replace the term "Same" with the periods, despite their redundancy.

7. Line 145. I would change "we use 24 week retweet windows" to "we use a time span of 24 weeks for our retweet networks"

8. Communities and hate speech, line 155: "elsewhere" is too colloquial. It should be more descriptive, briefly describing the context in which the methods described were used. Furthermore, given the importance of the claim, I suggest to cite a peer-reviewed work instead of an Arxiv one. My proposal is the following:

9. line 171: change "," to ", namely" or similar.

10. line 206: the comma should be replaced with a period since you are explaining two different standalone concepts.

11. line 225-227: sounds too mechanical to me. There is an overabundance of subordinates clauses.

12. Twitter users and hate speech, line 265: what does the term "Selected" refer to?

13. Conclusions, line 310-311: "consistent with the findings elsewhere" too colloquial and general.

14. line 312: I would change "difference" to "differences"

15. Hate speech classification, line 386: "elsewhere" is too colloquial. Please, give more context. Furthermore, the cited work lacks of a journal or conference name. Please provide more details about it.

Reviewer #2: The paper proposes an analysis of Online Social Media data based Twitter, studying a three-year continuum of Slovenian's Tweets in order to detect (using the state-of-the-art methodology) the hate speech on Twitter, with tweets classified as acceptable and unacceptable (with differentiation between inappropriate, offensive, and violent).

The authors also discuss the evolution of the network in order to understand, using social network analysis, where the hate speech comes from (political oriented accounts most), showing a slight increase in terms of share of the unacceptable tweets since the beginning of the data collection.

I want to make a few (minor) comments on the proposed work. I have also not corrected any grammatical mistakes.

L 121. The authors often differentiate between unacceptable and acceptable tweets, but provide no examples of each tweet (also for subcategories of the unacceptable tweets). Could the author provide, if possible, examples in a supplementary file?

I understand the authors want to summarize all methods and definition in one section after the main discussion, but, it may be helpful to have some (very short) explanation also in the result and discussion section. Sometimes they do that e.g. line 147, sometime they do not e.g. line 232 and line 241 (short explanation required).

L 265. The authors categorize the most important tweeters (890 users), but do the authors know the concentration amongst all users? In other words, could they also provide the users' concentration index, also independently from the classification of their tweets?

L 284 & L 316. Do the authors know how many profiles have been suspended by Twitter or closed over time?

L 334. The authors mention the subcategories of unacceptable tweets, it is unclear here if they are able to catch all the categories of tweets or if they are able to catch, consistently with the benchmark of the human annotators, a coherent (and consistent) amount of unacceptable tweets, if so, it would be helpful to know (e.g. extend the table 7) the classification statistics for each subclass of unacceptable, maybe in an supplementary appendix.

L 406. About the human annotator training, did authors share uniformly the dataset to the annotators(e.g., 5,000 tweets per annotator)?

L 417. I understand the use of the Krippendorff's alpha but, do the authors considered other measures - such as the Fleiss's k - to investigate the (dis)agreement between the raters?

L 458, could the authors give (beyond the reference cited) a brief definition of the approach (2 rows max), to clarify the methodology?

About citation #16 there is no reference available

6. PLOS authors have the option to publish the peer review history of their article (what does this mean?). If published, this will include your full peer review and any attached files.

Reviewer #1: No

Reviewer #2: No

---

## [Author Response · Author response to Decision Letter 0]

28 Jul 2021

We are submitting a revised manuscript titled “Retweet communities reveal the main sources of hate speech”. We considered all the points raised by the two reviewers. We really appreciate very detailed and constructive comments provided by both reviewers and would like to thank them. All the point-to-point answers to the reviewers’ comments are in the rebuttal.docx file.

---

## [Editor Report · Decision Letter 1]

20 Dec 2021

PONE-D-21-16242R1

Retweet communities reveal the main sources of hate speech

PLOS ONE

Dear Dr. Mozetič,

Thank you for submitting your manuscript to PLOS ONE. After careful consideration, we feel that it has merit but does not fully meet PLOS ONE’s publication criteria as it currently stands. Therefore, we invite you to submit a revised version of the manuscript that addresses the points raised during the review process.

We ask that you please address the following concerns:

1) Please provide further details in the Methods section on how hate speech, and the individual hate speech classes, are defined in this study.

2) Please clarify the criteria the annotators used to label the tweets into the different classes.

3) Please clarify how the communities were identified, and how the labels were chosen to describe the different communities identified.

4) Please clarify in your cover letter the changes made to the model on the link you have provided, and which problems you had identified that prompted this.

We look forward to receiving your revised manuscript.

Kind regards,

Hanna Landenmark

Senior Editor, PLOS ONE

on behalf of

Antonio Scala, PhD

Academic Editor

PLOS ONE
---

## [Author Response · Author response to Decision Letter 1]

22 Dec 2021

The response to the reviewers and editors is in the rebuttal letter, uploaded separately.

---

## [Editor Report · Decision Letter 2]

7 Mar 2022

Retweet communities reveal the main sources of hate speech

PONE-D-21-16242R2

Dear Dr. Mozetič,

We’re pleased to inform you that your manuscript has been judged scientifically suitable for publication and will be formally accepted for publication once it meets all outstanding technical requirements.

Kind regards,

Antonio Scala, PhD

Academic Editor

PLOS ONE
---

## [Editor Report · Acceptance letter]

9 Mar 2022

PONE-D-21-16242R2 

Retweet communities reveal the main sources of hate speech 

Dear Dr. Mozetič:

I'm pleased to inform you that your manuscript has been deemed suitable for publication in PLOS ONE. Congratulations! Your manuscript is now with our production department. 

Kind regards, 

on behalf of

Dr. Antonio Scala 

Academic Editor

PLOS ONE